# In-situ noncovalent interaction of ammonium ion enabled C–H bond functionalization of polyethylene glycols

Zongnan Zhang[1,4], Xueli Lv[2,4], Xin Mu[1], Mengyao Zhao[3], Sichang Wang[3], Congyu Ke[3], Shujiang Ding [1]✉, Dezhong Zhou [1]✉, Minyan Wang [2]✉ & Rong Zeng [1]✉

The noncovalent interactions of ammonium ion with multidentate oxygen-based host has never been reported as a reacting center in catalytic reactions. In this work, we report a reactivity enhancement process enabled by non-covalent interaction of ammonium ion, achieving the C–H functionalization of polyethylene glycols with acrylates by utilizing photoinduced co-catalysis of iridium and quinuclidine. A broad scope of alkenes can be tolerated without observing significant degradation. Moreover, this cyano-free condition respectively allows the incorporation of bioactive molecules and the PEGylation of dithiothreitol-treated bovine serum albumin, showing great potentials in drug delivery and protein modification. DFT calculations disclose that the formed α-carbon radical adjacent to oxygen-atom is reduced directly by iridium before acrylate addition. And preliminary mechanistic experiments reveal that the noncovalent interaction of PEG chain with the formed quinuclidinium species plays a unique role as a catalytic site by facilitating the proton transfer and ultimately enabling the transformation efficiently.

Inspired by nature, noncovalent interactions of ammonium ion with multidentate oxygen-based host, such as crown ether, have attracted considerable attentions in self-assembly processes[1–3]. Due to the good selectivity, high efficiency, and convenient responsiveness, this host-guest system has been widely used to construct functional molecular aggregates, such as artificial molecular machines[4], drug delivery materials[5], and supramolecular polymers[6], however, utilizations in catalysis[7] are still quite underdeveloped. Early works by Feringa[8], Leigh[9], and Leung[10], et al., revealed that noncovalent interactions of crown ethers with ammonium ion could switch reactivities of the well-designed catalysts to be on or off in nucleophilic addition or substituent reactions (Fig. 1A). Moreover, Costas, Olivo, Di Stefano[11] and Tiefenbacher[12] groups, et al, demonstrated that ammonium ion recognitions could enable remote C(sp3)–H oxidations of amines site-

selectively by using metal catalysts containing remote crown ether moieties (Fig. 1A). Although demonstrating critical roles as auxiliaries, these recognition processes are not true catalytic sites. And to the best of our knowledge, there have been no reports on the noncovalent interaction enabled reactivity enhancement in catalysis until now. During this process, noncovalent interactions might play a unique role as a key catalytic site by significantly enhancing the reactivity of ammonium ion and ultimately achieving the complete transformation efficiently.

On the other hand, the C–H functionalization of polyethylene glycols (PEGs) has become an increasingly useful tool since it could incorporate polar functional groups into the main chain in place of hydrogen to construct the multiply-functionalized PEGs, which are highly beneficial for many research fields including surface

[1]School of Chemistry & School of Chemical Engineering and Technology, Xi'an Jiaotong University, Xi'an 710049, P. R. China. [2]State Key Laboratory of Coordination Chemistry, School of Chemistry and Chemical Engineering, Nanjing University, Nanjing 210093, P. R. China. [3]College of Chemistry and Chemical Engineering, Xi'an Shiyou University, Xi'an 710065, P. R. China. [4]These authors contributed equally: Zongnan Zhang, Xueli Lv. ✉e-mail: dingsj@mail.xjtu.edu.cn; dezhong.zhou@xjtu.edu.cn; wangmy@nju.edu.cn; rongzeng@xjtu.edu.cn

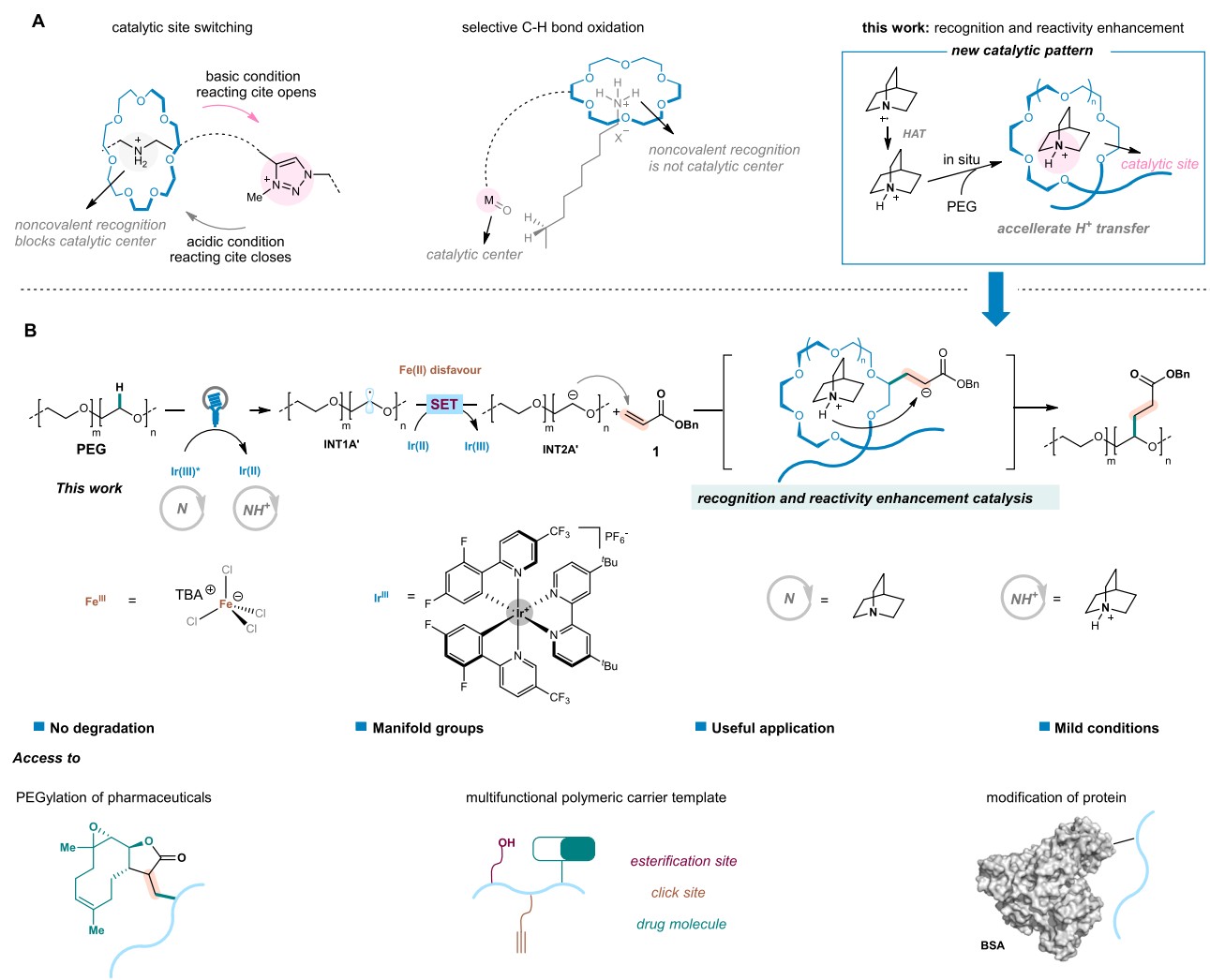

**Fig. 1 | Noncovalent interaction in catalysis. A** Noncovalent interactions of ammonium ion with multidentate oxygen-based host in catalysis. **B** Photoinduced C–H bond alkylation of PEGs with acrylates in this work.

modification[12–14], biofunctionalization[15–18], drug delivery[19,20], hydrogels[21,22], and solid-state electrolytes[23,24]. Among various scenarios, radical-based transformations are particularly attractive for PEGs functionalization due to the high reactivity under mild conditions. The early works by Parent[25], Elisseeff[26], and Bielawski[27]. et al., set the basis for designing radical-based hydrogen atom abstraction and substitution including alkenylation and oxidation. Nevertheless, owing to the severe competing chain scission via C–O cleavage and the lack of methods for substitution of C–H bonds in polymer, the mildly catalytic C–H functionalization of PEGs remains one of the most daunting fundamental and practical challenges. Impressive development in combination of hydrogen-atom transfer (HAT) and reversible addition–fragmentation chain transfer (RAFT) provided a powerful route for PEGs-initiated polymerization instead of simple C–H bonds substitution, reported by Fors group[28]. Until recently, the selective incorporation of polar groups into the PEG chains without significant chain scission was reported for the first time via the photoinduced iron-catalyzed alkylation of polyethers with polar olefins[29]. While the electron transfer processes of the iron catalyst via the ligand-to-metal charge transfer (LMCT) and carbon radical reduction were crucial[30–32], however, the relatively low reduction potential of Fe(II)/Fe(III) (−0.38 V $vs$ Fc$^{0/+}$, Supplementary Fig. 46) resulted in the limited scope of alkenes since only di(electron-deficient group)-substituted alkenes could be utilized (Fig. 1). Particularly, while the existence of cyano group might

arise the concern on the potential biological toxicity[33], the nitrile-free alkenes presented significantly lower reactivity. To meet the requirements in the specific application, such as drug delivery[19,20], additional steps are still typically required. Despite the advances realized, a more general strategy for C–H functionalization of PEGs, particularly alkylation using acrylates, is still an uncharted cartography.

Given that the dual catalysis of polypyridyl iridium complex and organic Brønsted base, such as quinuclidine, under visible light could facilitate the hydrogen abstraction[34–37] and present a higher reduction potential (Ir[dF(CF$_3$)ppy]$_2$(dtbbpy)PF$_6$ **PC-1**, Ir(II)/Ir(III), −1.26 V $vs$ Fc$^{0/+}$, Supplementary Fig. 47), we wondered whether such a dually catalytic system would be possible to achieve the coupling reaction of PEGs with acrylates via an electron transfer process. If successful, we anticipated that such a strategy might constitute a worthwhile endeavor for chemical invention, offering an unrecognized opportunity to efficiently synthesize functionalized PEGs. However, at the outset of our investigations, it was still unclear whether it could be designed given (a) Ir/quinuclidine catalysis not being used in polymer functionalization, (b) strong coordination between cationic Ir complex with the multidentated PEGs chain, (c) chain entanglement of PEGs preventing interaction between two catalysts, (d) dioxane as a model substrate failed being functionalized with acrylates under Ir/quinuclidine dual catalysis.

Herein, we report a visible-light-promoted co-catalysis platform of iridium complex and quinuclidine to accomplish an example on

controllable C−H bond functionalization of PEGs with acylates without significant polymer chain scission. A reaction sequence involving the generation and direct reduction of $\alpha$-carbon radical adjacent to oxygen atom and subsequent nucleophilic addition is proposed based on the DFT calculations. Moreover, the spontaneous noncovalent interaction of PEG chain with the formed ammonium ion in situ is crucial[1,38], revealing an unexpected mechanism for the transformation (Fig. 1B). This is also a noncovalent interaction of ammonium ion enabled reactivity enhancement catalysis. A series of drug molecules and even a protein could be incorporated into the main chain of PEGs through this efficient sequence, presenting a diversely worthwhile protocol for drug delivery.

## Results

### Discovery and optimization

We began our study by evaluating the reactivity of PEG 2000 with benzyl acrylate **1** (20 mol%) using a combination of Ir[dF(CF$_3$) ppy]$_2$(dtbbpy)PF$_6$ **PC-1** (0.1 mol%) and quinuclidine (2 mol%) as dual catalysts in acetonitrile (MeCN, 0.5 M) under irradiation with 460 nm LEDs for 12 h[24]. We were pleased to find that these conditions failed to enable the reaction of dioxane with benzyl acrylate **1**, however, allowed efficient C(sp$^3$)−H alkylation of PEG 2000 to occur, obtaining the desired product **PEG 2000-1** with 8.3 mol% of level of functionalization (LOF) (entry 1, Supplementary Table 1), suggesting a different mechanism be involved. We subsequently studied the influence of the catalysts and found that both **PC-1** and quinuclidine were indispensable, since the reactions did not happen in the absence of either **PC-1** or quinuclidine (entries 2-3). Other photo-catalysts, such as Ir(ppy)$_2$(dtbbpy)PF$_6$ (**PC-2**, entry 4), [Mes-Acr]BF$_4$ (**PC-4**, entry 6)[39], and 4-CzIPN (**PC-5**, entry 7)[40] did not give the product, while Ir[dF(CF$_3$) ppy]$_2$(bpy)PF$_6$ (**PC-3**, entry 5) observed lower efficiency with 6.0 mol% of LOF. Other photosensitive hydrogen abstraction catalysts, such as TBAFeCl$_4$ and benzophenone, were proven to be inefficient ones (entries 8-10). Switching the solvent to DCM and DMF shut down the reaction (entries 11, 13) and acetone led to somewhat lower efficiency (entry 12).

The time-dependent experiments were then conducted to understand the relationship between level of functionalization (LOF) and reaction time. When 20 mol% of alkenes **1** was used, the LOF values are highly dependent on the reaction times. Along with increase of reaction time, the LOF was determined to be 8.3 mol% in 12 h, and finally reached the high LOF as 17.6 mol% in 48 h (Fig. 2A). The kinetic experiments revealed that the reaction rate highly relied on the concentration of PEG and acrylate instead of those of Ir and quinuclidine catalyst (Supplementary Figs. 4–7). The molecular weight distribution (MWD) of the products, including $M_w$, $M_n$, and Đ values, were then analyzed using gel permeation chromatography (GPC) (Fig. 2B). Not surprisingly, the $M_w$ and $M_n$ persistently increased along with increase of reaction times from 0 to 48 h, while the Đ values remained relatively low as approximate 1.10 (Fig. S3). Importantly, the core property of polyether chain could be remained well since no significant chain scission caused by $\beta$-O elimination was observed. Moreover, the yielded product **PEG 2000-1** was sequentially good-characterized using FT-IR, $^1$H NMR, $^1$H-$^1$H COSY, heteronuclear multiple bond correlation (HMBC), and particularly diffusion ordered NMR spectroscopy (DOSY) spectra, identifying the incorporation of the small molecule into polymer chain be covalent bonding instead of noncovalent mixing (Fig. 2C)[41]. In addition, although the matrix-assisted laser desorption/ionization time-of-flight (MALDI-TOF) mass spectrum of **PEG 2000-1** was measured but failed to gain the acceptable information, the corresponding spectra of PEG 2000 and product **PEG 2000-2** (LOF = 4.8 mol%) formed from methyl acrylate **2** were successfully collected and analyzed, confirming the incorporation of polar alkenes into the polymer chain. The molecular weight (MW) increase was identified as about two molecules of alkene **2** installing in PEG$_{45}$ chain (PEG 2000),

which accounted for 4.4% of LOF and consistent with $^1$H NMR analysis (LOF = 4.8 mol%).

### Scope and applications

With the optimal condition in hand, the scope of acrylates was next evaluated using PEG 2000 and 20 mol% of different acrylates. In the presence of **PC-1** (0.1 mol%) and quinuclidine (2.0 mol%) under blue light (460 nm) for 5 h, the reaction of PEG 2000 (Đ = 1.10) with acrylate **1** and **2** proceeded smoothly to obtain desired products **PEG 2000-1 - 2** with 5.5 mol% and 4.8 mol% of LOF, respectively. The GPC curves maintained narrow to give low Đ values. The substrates containing functional groups, such as -CF$_3$ (**3**), -Cl (**4**), -Br (**5**), terminal alkynyl (**6**), and hydroxy group (**7**), all led to the desired modified PEGs with good LOF (3.2-9.3 mol%) and narrow Đ values (1.08-1.12), showing great potential for further conversion in biorthogonal chemistry[42–44]. Besides acrylates, other electron-deficient alkenes, such as *N*-phenylacrylamide **8**, phenyl vinyl sulfone **9**, and diethyl vinylphosphonate **10**, could be compatible under this catalytic system. While alkenes with higher steric hindrance, such as methyl methacrylate **11** and chiral dehydroalanine **12**, were used, the C−H bond functionalization reactions were still proceeded smoothly to furnish the desired products **PEG 2000-11** and **PEG 2000-12** with great LOFs (6.3 mol% and 7.0 mol%, respectively) and Đ values (1.11 and 1.09, respectively) (Fig. 3).

Based on the successful results of acrylates scope, we attempted to utilize this strategy to accomplish one-step PEGylation of bioactive molecules via C−H bond functionalization (Fig. 3), which might improve stability and pharmacokinetic properties of drugs and constitute a worthwhile endeavor for drug delivery[19,20]. The reactions of PEG 12,000 with 10 mol% of drug-containing acrylamide derivatives, such as ibrutinib **13** and desloratadine **14**, under iridium-quinuclidine dual catalysis were firstly conducted to obtain the PEGylation products successfully with 1.2 mol% and 2.9 mol% of LOF, respectively. A bit amount of polymer cross-linking products might be generated during PEGylation since a high molecule shoulder was observed in GPC curves. Switching the acrylamide derivatives to acrylates, we next examined the reactions using five-member lactone-containing drug molecules, such as parthenolide **15** and isoalantolactone **16**. Meanwhile, these PEGylation reaction proceeded smoothly without observing significant cross-linking of polymer chain in the GPC traces. Another efficient strategy for PEGylation of drug molecules could be developed through a two-step protocol involving the incorporation of 2-hydroxyethyl methacrylate **17** and the subsequent condensation with carboxylic acid containing drug molecules. A series of bioactive molecules could be PEGylated. A gram-scale product of **PEG 12000-17** was firstly obtained with 3.4 mol% of LOF and low Đ value 1.12. The esterification reactions of **PEG 12000-17** with 2 mol% probenecid **18**, indomethacin **19**, or lithocholic acid **20**, were then proceeded to form the PEGylation products, remaining low PDI (polydispersity index) values of PEG polymer (1.10-1.11).

Moreover, this efficient strategy on C−H bond functionalization of PEG further allowed the modular synthesis of the multi-functionalized polymer **PEG 12000-17-6-15** via the cascaded incorporation of diverse functional groups using different electron-deficient alkenes[42]. The hydroxy and terminal alkynyl groups were firstly introduced into PEG chain sequentially by reacting with 2-hydroxyethyl methacrylate **17** and prop-2-yn-1-yl acrylate **6**. The LOFs of hydroxy group and terminal alkyne were determined to be 2.4 mol% and 4.1 mol%, respectively. Finally, the drug molecule parthenolide **15** could be installed to form polymer **PEG 12000-17-6-15**. The existence of a drug molecule and the convertible hydroxy and terminal alkynyl groups provided a great platform for further transformations, such as esterification, coupling reaction, and click reaction, to link with other potential application sites (Fig. 4A).

In addition, since PEGylation of proteins had been widely utilized in medical fields such as prolonging blood circulation time[15], a strategy

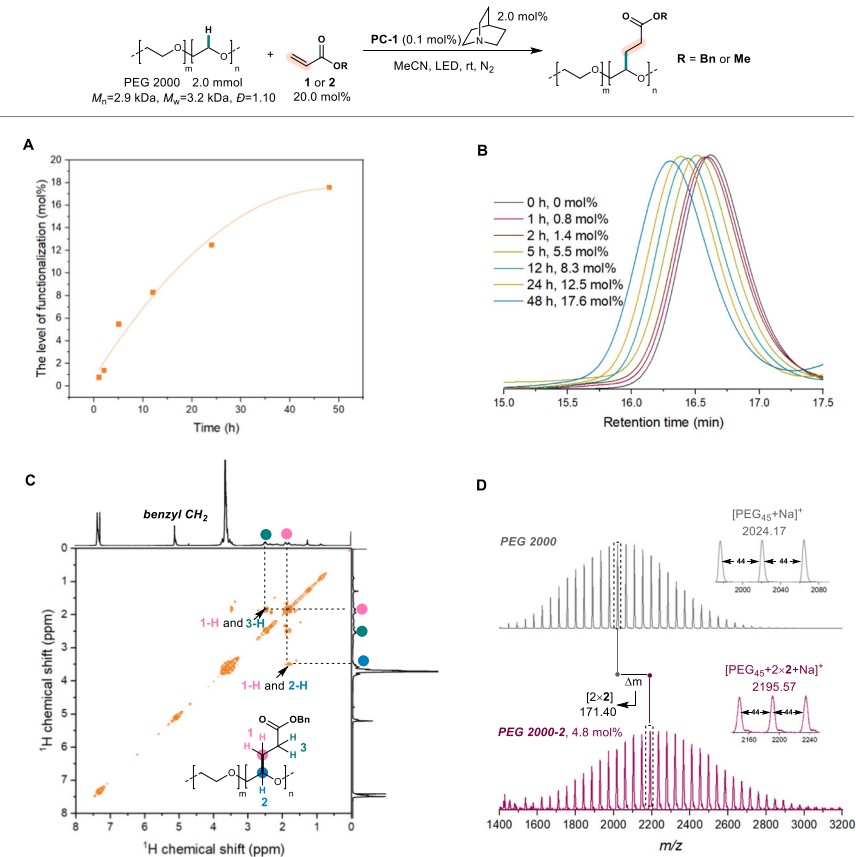

**Fig. 2 | C − H alkylation of PEG 2000 and acrylates. A** LOFs of PEG 2000 with **1** vs reaction time. **B** GPC curves of **PEG 2000-1** with various LOFs. **C** ¹H-¹H COSY spectrum of **PEG 2000-1**. **D** The MALDI-TOF mass spectra of PEG-2000 and **PEG 2000-2**.

involving C−H bond functionalization of PEG was designed and realized. Firstly, **MeO-PEG-OMe 2000-17** was prepared successfully using the photoinduced iridium/quinuclidine catalysis for 3 h under blue light with 2.2 mol% of LOF and 1.12 of Đ value. The esterification of **MeO-PEG-OMe 2000-17** with 3-maleimidopropionic acid was then proceeded to introduce the 1H-pyrrole-2,5-dione moiety, which underwent Click reaction with mercapto group at the protein, such as the dithiothreitol (DTT)-treated BSA (bovine serum albumin) with free -SH groups, in phosphate buffer (pH 8.0) at room temperature for 2 h. The PEG-protein conjugate was obtained successfully, which could be detected by sodium dodecyl sulfate-polyacrylamide gel electrophoresis (SDS-PAGE) (Fig. 4B)[34].

## Mechanism

The mechanism for the dually Ir/quinuclidine-catalyzed C−H bond alkylation of PEG was next explored. First, the comparable control experiments using Ir or Fe catalyst and different substrates were examined (Fig. 5A). As reported[29], TBAFeCl₄ catalyst was efficient for 2-benzylidenemalononitrile **22** but not acylates to react with the model small molecule (dioxane) and PEGs due to the relatively low reduction potential of Fe(II)/Fe(III) (−0.38 V vs Fc⁰/⁺). As a comparison, when Ir complex and quinuclidine were used, the higher reduction potential of Ir[dF(CF₃)ppy]₂(dtbbpy)PF₆ (Ir(II)/Ir(III), −1.26 V vs Fc⁰/⁺) indeed enabled the reduction of the involved carbon radical and facilitated the reaction of PEG with acrylates, however, the reactions of 2-benzylidenemalononitrile did not proceed with either dioxane or PEG, which was quite distinct. In addition, a huge difference in reactivity with benzyl acrylate **1** between small molecules and polymer was unexpectedly observed. PEG could react with **1** to form the desired functionalized polymer, nevertheless, the model small molecules, such as dioxane and dimethoxyethane (DME), showed extremely low

reactivities (trace and 13%, respectively). These results had arisen three considerable concerns on (1) different reactivities between Ir/quinuclidine and iron catalysis using acrylates, (2) Ir/quinuclidine catalysis being incompatible with 2-benzylidenemalononitrile, (3) PEGs being tolerated well but small molecule, 1,4-dioxane and DME, not.

To address the first concern, the detailed density functional theory (DFT) calculations were conducted to explore the processes of free radical generation and quenching, as well as the differences between iridium/quinuclidine and iron catalytic systems[45–47]. The catalytic cycle obtained by computational calculations using 2,5,8,11,14,17-hexaoxaoctadecane (named as **PEG260**) and benzyl acrylate **1** as the model reaction is shown in Fig. 5B. Under blue light excitation (460 nm), the photocatalyst **Ir(III)** is excited to its photoexcited state *****Ir(III)** with an endothermal energy of 57.1 kcal·mol⁻¹. Subsequently, the intermediate *****Ir(III)** undergoes reductive quenching through a single electron transfer (SET) process with quinuclidine as the oxidizing agent. This results in the formation of **Ir(II)** and simultaneous oxidation of quinuclidine, generating the quinuclidine radical. The quinuclidine radical abstracts a hydrogen atom from **PEG260** through transition state **TS1A** with an energy barrier of 12.8 kcal·mol⁻¹, resulting in the formation of radical species **INT1A**. The hydrogen bond weak interactions are observed in **INT1A**. The transformation of **INT1A** might involve two competing pathways. Intermediate **INT1A** is directly reduced by **Ir(II)** to form **INT2A** and regenerate the photocatalyst **Ir(III)**. Then, **INT2A** undergoes a Michael-type nucleophilic addition with the carbon-carbon double bond of substrate **1** through transition state **TS2A**, with an energy barrier of 8.0 kcal·mol⁻¹. Alternatively, a Giese-type radical-addition pathway is also considered. In this pathway, the radical **INT1A** is captured by acrylic ester **1**, leading to the formation of **INT2B** through transition state **TS2B**, which is then reduced by **Ir(II)**. Both pathways ultimately result in the formation of **INT3A**. The frontier

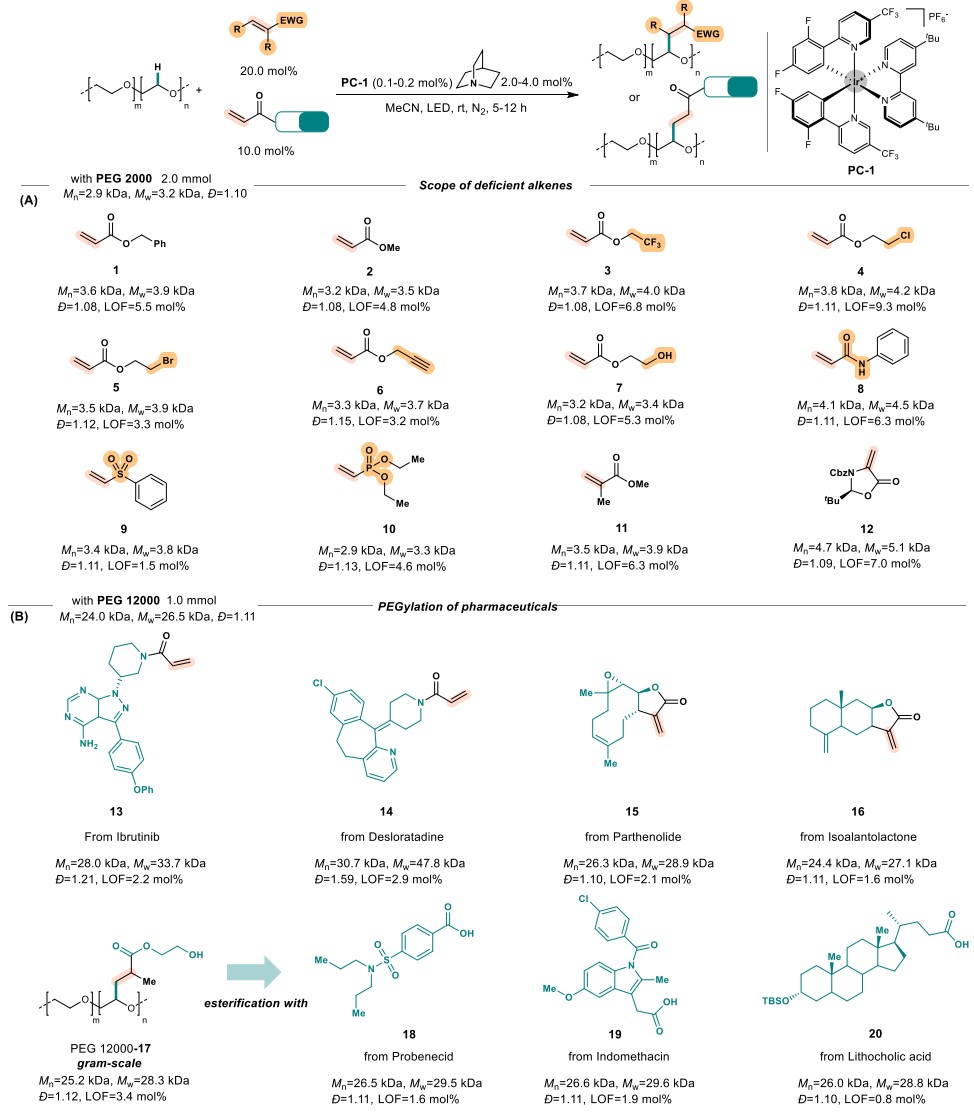

**Fig. 3 | Reaction scope. A** Scope of electron-deficient alkenes. **B** PEGylation of bioactive molecules.

molecular orbital was then used to analyze the feasibility of the two pathways mentioned above (Fig. 5C). The singly occupied molecular orbital (SOMO) of intermediate **Ir(II)** donates an electron to the lower energy level SUMO (singly unoccupied molecular orbital) orbital of the radical species, enabling a single electron reduction. Meanwhile, the SUMO orbital of the radical species accepts the electron and was reduced into a negative ion. The calculated energy data suggests that, the 1e-reduction process of **INT1A-radical** and **INT2B-radical** by **Ir(II)** is thermodynamically permissible. Notably, when substrate **22** acts as a Michael acceptor, the strong electron-withdrawing effect of the cyano group significantly lowers the energy of the SUMO orbital of **INT2B-22-radical**. In the photoinduced iron-catalyzed system, the excited state $^7$**Fe(II)*** can only reduce **INT2B-22-radical**, and cannot achieve catalytic cycles in reactions involving acrylates. We further investigated the 1,2-Michael nucleophilic addition of intermediate as a competing reaction. The energy barrier of transition state **TS2C** is 14.7 kcal·mol⁻¹, which is much higher than that of **TS2A** for 1,4-Michael nucleophilic addition (14.7 kcal mol⁻¹ vs 8.0 kcal mol⁻¹, Fig. 5B). The generated **INT3A** undergo protonation to produce the desired product. This process is accompanied by the regeneration of quinoline, completing the catalytic cycle of the base. Compared with **PEG260**, we conducted separate analyses on the crucial transition states of dioxane and 1,2-dimethoxyethane (DME) in the Ir-catalyzed cycle of their reactions with

acrylates, as illustrated in Fig. 5D. The energy barrier observed for the transition state **TS1A-dioxane** in hydrogen atom transfer (HAT) process is significantly higher compared to those of DME and **PEG260** (21.0 vs 14.0, 12.8 kcal·mol⁻¹). This high energy barrier limits the cleavage of carbon-hydrogen bonds in dioxane, which hinders the progression of the reaction. For substrate DME, the energy barrier for 1,4-addition through **TS2A-DME** is comparable to the 1,4-nucleophilic addition through **TS2C-DME** (Fig. 5B). As a result, the observed yield of the reaction between DME and acrylates is only 13% in yield. Increasing the chain length of ethers enhances the stability of the transition state, resulting in a much larger energy decrease for the nucleophilic 1,4-addition compared to the 1,2-addition. This leads to the predominance of nucleophilic 1,4-addition of polyethylene glycols as the main alkylation product, which is consistent with the experimental observations.

The pK$_a$ values of the potential intermediates were next compared to address the second concern (Fig. 6A). While it was known that pK$_a$ values of [quinuclidine-H]⁺, 2-alkylmalononitrile, and ester in DMSO are 9.8[48], -12.5[49], -22.7[50], respectively, the strong basicity of the carbanion adjacent to the carbonyl group would be responsible for the rapid protonation with the weak acid [quinuclidine-H]⁺ (pK$_a$ = 9.8), driving the complete conversion of the acrylate (Fig. 6B). In contrast, the relatively lower basicity of the carbanion from 2-alkylmalononitrile might lead to the slower protonation, resulting in the low reactivity of

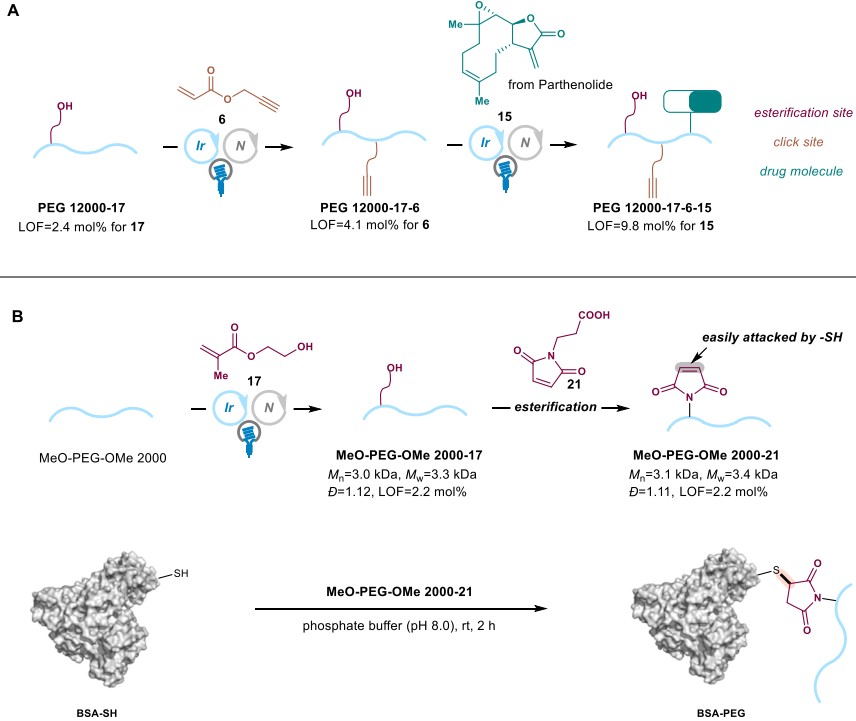

**Fig. 4 | Application of C–H bond functionalization of PEG. A** Modular synthesis of multifunctional polymer **PEG 12000-17-6-15**. **B** PEGylation of BSA.

2-benzylidenemalononitrile under Ir/quinuclidine catalysis (Fig. 6B). The results using iron catalysis could be also explained since the strong acid (HCl, pK$_a$ = 1.8)[51] instead of [quinuclidine-H]$^+$ was generated in situ to favorably protonate carbanion adjacent to nitrile groups.

For the reason why PEGs were well-tolerated but 1,4-dioxane not, inspired by host-guest recognition of organic ammonium cations with crown ethers, we firstly envisioned that the supramolecular interaction of [quinuclidine-H]$^+$ and PEG chain might increase the acidity of [quinuclidine-H]$^+$ species[1,38]. A series of experiments were then conducted to examine such a hypothesis (Supplementary Figs. 37–45). [Quinuclidine-H]$^+$(OTFA)$^-$ was then prepared by mixing quinuclidine with trafluoroacetic acid, whose pH value in MeCN was measured to be 2.09 ± 0.03. Interestingly, while PEG 2000 solely represented pH as 9.80 ± 0.04, the mixture of PEG 2000 and [quinuclidine-H]$^+$(OTFA)$^-$ obtained a pH as 1.84 ± 0.04, indicating that PEG 2000 chain could indeed entangle the cation via supramolecular interaction and then enhance the corresponding acidity (Fig. 6C). As comparisons, the small ethers, such as dioxane and DME, did not increase the acidity. This conclusion could be further confirmed by the analysis of the $^1$H NMR chemical shifts of N$^+$–H species, since [quinuclidine-H]$^+$ in the presence of PEG 2000 presented a significant down-field change (11.8 ppm) comparing with that of pure salt (10.9 ppm) (Supplementary Fig. 38). The interaction between PEG 2000 and [quinuclidine-H]$^+$ might cause pH value decrease, however, this acidity increase conclusion could further be ruled out by the experimental results of the C–H bond alkylation of crown ethers with different ring size from 12-crown-4 to 24-crown-8. When crown ethers were mixed with [quinuclidine-H]$^+$(OTFA)$^-$ and conducted the pH value examination, 15-crown-5, 21-crown-7, and 24-crown-8 maintained or increased the pH values to be 2.11 ± 0.03, 3.62 ± 0.02, 3.57 ± 0.02, respectively, while 18-crown-6 could decrease it to 1.72 ± 0.03, which was very close to that of PEG 2000 (Fig. 6C). Paradoxically, the C–H bond alkylation of crown ethers all observed promising results, and the reactivities increase along with the oxygen-containing numbers of substrates (Fig. 6D).

The reactions of glymes **G-1** to **G-4** with acrylate **1** were then explore and obtained similar results since the reactivities of glymes were also positively related to the oxygen-containing numbers (Fig. 6E). These results indicate that the noncovalent interaction of [quinuclidine-H]$^+$ with oxygen-containing poly-dentated substrates is probably highly responed to the stability of [quinuclidine-H]$^+$ and the subsequent proton transfer, which results in reactivities enhancement. Notably, this conclusion could be also supported by the results on gas phase protonation of polyethers, glemes, and crown ethers, seminally reported by Kebarle in 1984[52].

Based on the above experiments, a plausible mechanism has been proposed to explain the interaction and reactivity. Firstly, the C–H bond abstraction of PEG with quinuclidine cationic radical via HAT pathway could generate carbon radical and quinuclidine-H$^+$ species. The interaction between quinuclidine-H$^+$ and poly-coordination of PEG chain stabilizes the proton and enables the formation of **INT2**. The single electron reduction with photocatalyst and subsequent nucleophilic addition with acrylate **1** can produce **INT3**, which undergoes a "formal intramolecular" proton transfer to yield the final product. (Fig. 6F).

## Discussion

In summary, we have utilized the noncovalent interaction and photoinduced co-catalysis of iridium and quinuclidine to develop the C–H bond alkylation of PEGs with acrylates without observing signification degradation. This is also a noncovalent interaction of ammonium ion enabled reactivity enhancement in catalysis. A series of functional groups, such as halogen, terminal alkyne, hydroxy, sulfonyl, and phosphate groups et al, in the acrylate derivatives did not affect the efficiencies significantly, presenting a universally powerful tool for C–H bond alkylation of PEGs. This method was also able to respectively realize the corporation of the bioactive molecules, multi-functionalization of PEG, and PEGylation of DTT (dithiothreitol)-treated BSA (bovine serum albumin) protein through an efficient reaction sequence. Moreover, based on the DFT calculations and plausible mechanistic experiments, a mechanism involving the hydrogen atom abstract, direct reduction of carbon radical, nucleophilic addition, and subsequent protonation was disclosed. Particularly, the intermediate formed from the in situ noncovalent

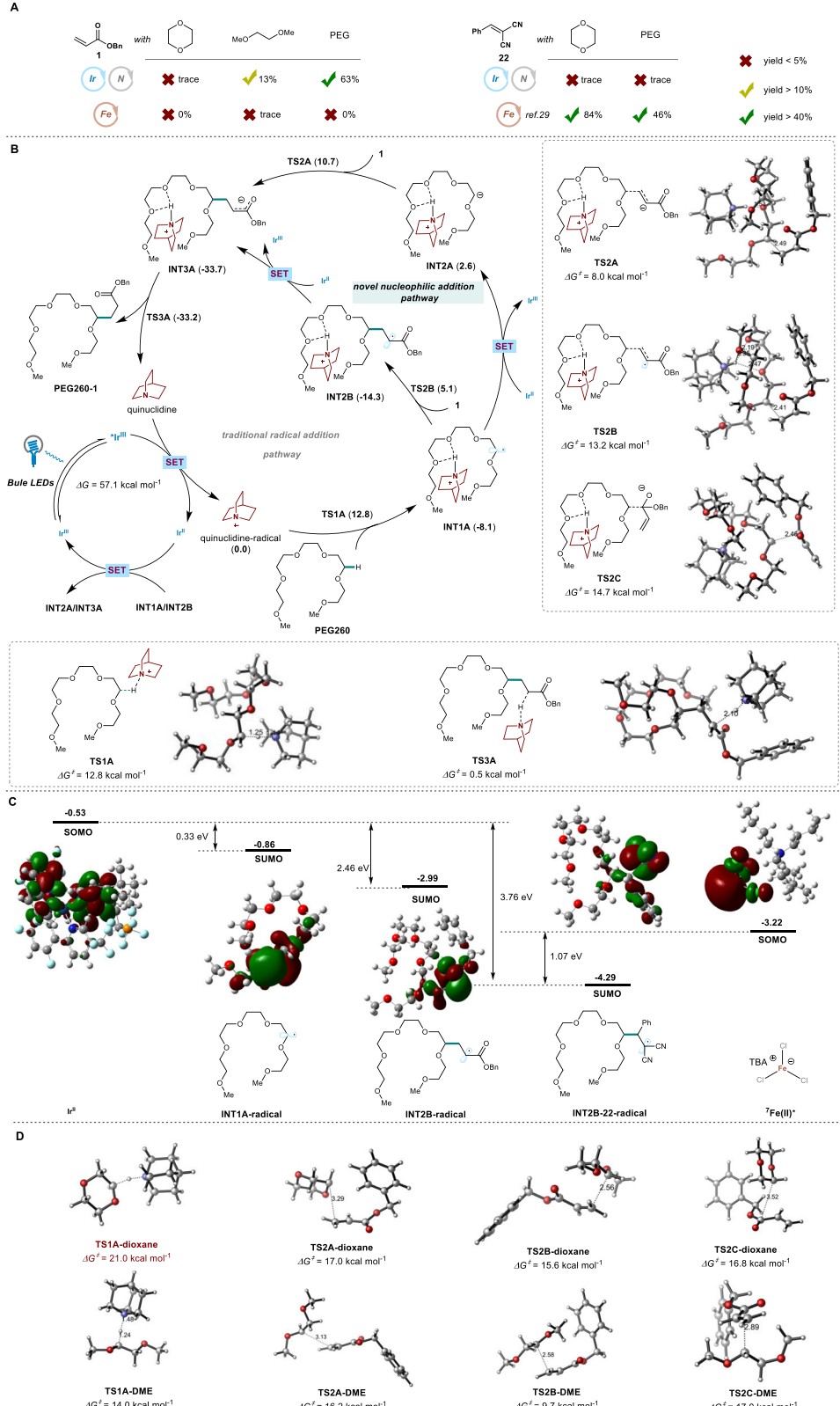

**Fig. 5 | Control experiments and DFT calculations. A** Control experiments of acrylate **1** or benzylidenemalononitrile with dioxane, DME, and PEG using iridium or iron catalysis. **B** Proposed mechanism for the photo-induced C−H Bond functionalization. **C** Frontier molecular orbital analysis. **D** The key transition states of dioxane and diglyme in the Ir-catalyzed cycle.

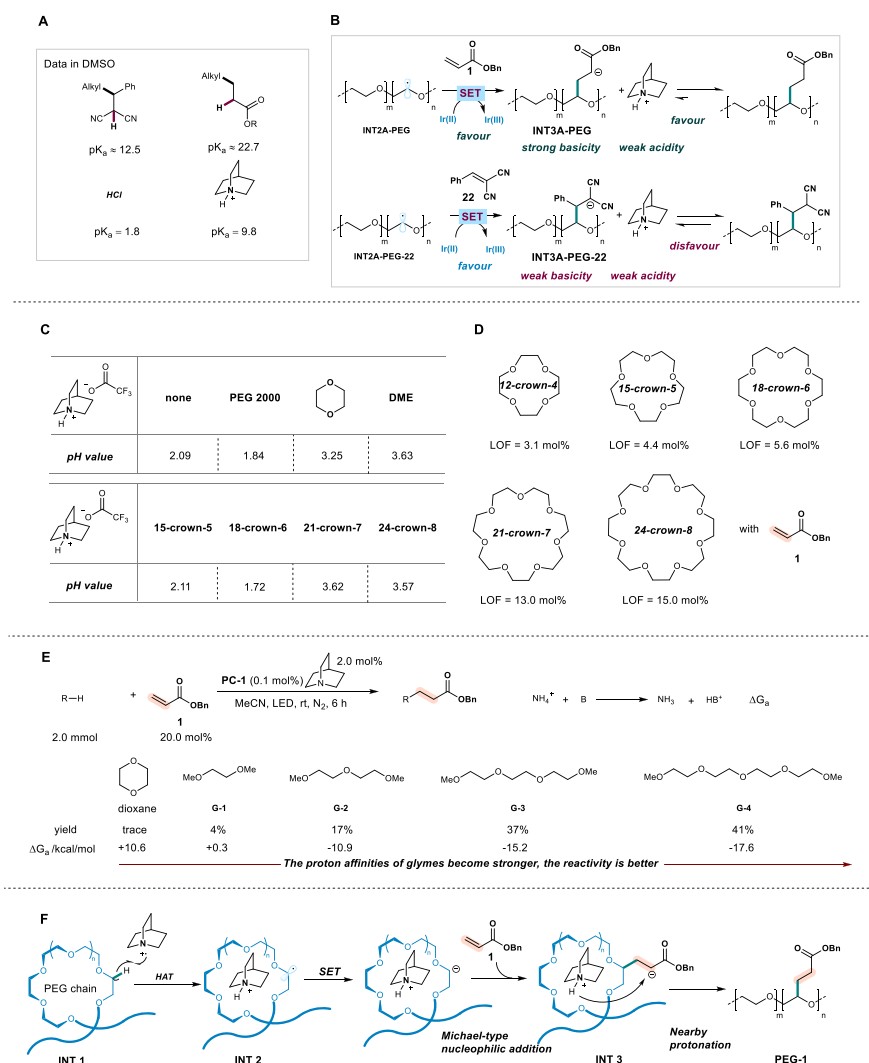

**Fig. 6 | Mechanistic experiments. A** pKa values of the potential intermediates. **B** The explanation on the selectivity of iridium catalysis. **C** pH values of quinuclidine trifluoroacetic acid salt in the presence of diverse additives. **D** The reactivity of 12-crown-4, 15-crown-5, 18-crown-6, 21-crown-7, and 24-crown-8 with acrylate **1**. **E** The reactivity of dioxane and glymes with acrylate **1**. **F** A plausible mechanism involving the supramolecular interaction enabled proton exchange.

interaction of PEG chain with quinuclidinium species was proven to be crucial to stabilize the cation species, enable proton exchange, and facilitate the transformation. The successful development of this method illustrates the great possibility of C−H functionalization of polymers and host-guest chemistry and enlightens a series of applications in many research fields.

## Methods

### A procedure for synthesis of PEG 2000-3

To a 4 mL vial were added **PC-1** (2.4 mg, 0.002 mmol), quinuclidine (4.5 mg, 0.04 mmol), PEG 2000 (88.5 mg, 2.0 mmol), **3** (52 μL, $d = 1.216$ g/mL, 61.6 mg, 0.4 mmol), and MeCN (4.0 mL) in an $N_2$ glovebox. The vial was then sealed and transferred out of the glovebox. Under irradiation at 460 nm LEDs, the resulting mixture was stirred for 5 h at rt. Evaporation and Flash chromatography on silica gel (DCM to DCM/MeOH = 10/1) afforded **PEG 2000-3**.

## Data availability

All the data supporting the findings of this study are available within the article and its Supplementary Information, or from the corresponding author on request. Source data of the DFT calculations are provided with this paper. Source data are provided with this paper.

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

## Acknowledgements

R.Z. is grateful for the financial support from the National Natural Science Foundation of China (22371223) and the Xiaomi Young Talents Program. M.W. is grateful for the financial support from the National Key R&D Program of China (2022YFA1503200). We thank Prof. Shiyong Liu, Prof. Jinming Hu, Mr. Jie Xu and Dr. Qiangqiang Shi from the University of Science and Technology of China for assistance with MALDI-TOF analysis, and Dr. Chao Feng from the Instrument Analysis Center of Xi'an Jiaotong University for assistance with NMR analysis. We are grateful to the High-Performance Computing Center of Nanjing University for

performing the numerical calculations in this paper on its blade cluster system.

## Author contributions

R.Z. and M.W. supervised the project. R.Z. and Z.Z. designed the experiments. Z.Z., X.M., M.Z., S.W. and C.K. performed and analyzed the experiments. Z.Z. and D.Z. performed and analyzed the GPC spectra. X.L and M.W. performed the DFT calculation. Z.Z., S.D., M.W. and R.Z. prepared this manuscript.

## Competing interests

The authors declare no competing interests.
