## [Peer Review File · Nature Communications]

In-situ Noncovalent Interaction of Ammonium Ion Enabled C–H Bond Functionalization of Polyethylene GlycolsReviewers' Comments:

Reviewer #1:

Remarks to the Author:

This communication deals with the functionalization of polyethylene glycol with a series of acrylate substrates by a photocatalytic process involving an iridium photocatalyst and quinuclidine. The synthetic results obtained and described in Table 1 are interesting but the manuscript has a number of severe issues that make it unpublishable in the present form. Here below, I will list some of these problems in order to be as constructive as possible for the authors.

1) First of all the language. I don't speak about English, that can be improve, but about the tones used by the authors. A research publication is not a publicity flyer. Along the manuscript one finds sentences in italics like "...the first example on controllable....." or "...first example on the ammonium ion recognition...". Every time that an interesting result is published, it is the first time it is observed, otherwise it is not so interesting. There is no needing to sell anything, thus more moderate tones are required. Also term as "excellent" should be avoided as well as "we rigorously characterized.....". A rigorous characterization is taken for granted.

2) While the synthetic part of the paper has a certain degree of interest and may be published in a more specialized journal, the second part of the paper, which occupies more or less half of the manuscript is really wanting. The second section is intitled Mechanism. The study of a reaction mechanism necessary has to involve kinetic measurements. In silico chemistry with imposed reaction trajectories may serve to explain experimental results and cannot substitute an experimental investigation based on kinetic measurements (kinetic constants for a related kinetic scheme). I see that, in the last period, such a publication scheme (mechanisms only based on in silico calculations) is becoming a fashion, but it doesn't bring anywhere.

3) I do not see any really stabilizing interaction between 18C6 (and PEG) and quinuclidinium ion. The authors cite ref 37 to illustrate the binding between tertiary ammonium ions and crown ethers. In the cited article by Izatt and co-workers, in Table 1, concerning the binding reaction between quinuclidinium iodide (compound 5) and 18C6 in methanol, I read (note d):

"Heat produced in these reactions is so small that logK and ΔH cannot be calculated, leading to the conclusion that ΔH and / or logK is very small"

From my personal experience, even dialkylammonium ion have very low affinity for 18C6. Thus, the argument presented at page 18, based on the pH of different solutions of quinuclidinium trifluoroacetate in the presence of the different ethers is too weak to demonstrate the point.

Overall, although the results described in the first part of the manuscript have a discreet synthetic interest, my view is that the manuscript as a whole is not publishable in the present form.

Reviewer #2:

Remarks to the Author:

In this manuscript, the authors successfully established an interesting method to realize C-H bond alkylation of PEGs by co-catalysis of iridium and quinuclidine without observing their signification degradation. A series of acrylate derivatives were suitable for this postmodification. Based on the DFT calculations and detailed mechanistic experiments, the mechanism involving the hydrogen atom abstract, direct reduction of carbon radical, nucleophilic addition, and subsequent protonation was disclosed. This work will provide an alternative for PEGs modification and expand their applications in divers areas. The manuscript is recommended acceptance after addressing following issues.

1. Has the equivalence ratio of the catalyst been optimized? Because dual catalysis is used in this work, the proportion of catalyst dosage is mostly likely to affect the reaction efficiency.
2. Although a series of acrylate derivatives were attempted, the conversion rate was not considered high. Based on the speculated mechanism, it is most likely that the efficiency of photocatalytic redox is insufficient. Please provide a reasonable explanation and prove that the catalytic system used in the work is sufficiently successful.

3. The structure of crown ethers will facilitate the Michael addition of acrylates. However, the formation of PEG crown ethers will be destroyed gradually with the increase of the level of functionalization, and the reaction efficiency will become worse and worse, which is an undeniable reason for limiting C–H bond alkylation of PEGs. The authors can further improve the relevant catalytic mechanism, perhaps to obtain better reaction results.
4. The energy calculations for INT1A and INT2B seem to be incorrect. Based on the general understanding, ester carbonyl groups can be conjugated with carbon radicals to stabilize molecular configurations, therefore INT2B should have lower molecular energy.
5. Lines 325-330 of page 17, the authors claimed that "In contrast, the relatively lower basicity of the carbanion from 2-alkylmalononitrile might lead to the slower protonation, to favorably protonate carbanion adjacent to nitrile groups." Can the Michael addition of 2-benzylidenemalononitrile be achieved by replacing quinuclidine with a less alkaline tertiary amine? Please provide reasonable analysis and reasons.

Reviewer #3:

Remarks to the Author:

The authors reported C-H functionalization of PEGs, which is enabled by in-situ noncovalent interaction of ammonium ion. This work is based on their recent publication (Controllable C–H Alkylation of Polyethers via Iron Photocatalysis, *J. Am. Chem. Soc.* 2023, 145, 13, 7612–7620), but presented much broader substrate scope and a new catalytic system. This may be a significant breakthrough, but the novelty of this work should be emphasized with greater clarity. It gave me the impression that the work mostly borrowed the catalyst system from literature, including the work of Macmillan, etc (see refs 33-36). The main difference is that the authors used PEG, instead of small molecules, in this work. The introduction as well as the results and discussion should be significantly revised accordingly. This work might be qualified, but I don't see such novelty and significance for publication on *Nature Communications* at this stage.

The characterizations of polymer products are not enough, which might lead to compromised conclusions. The authors used the product PEG 2000-1 as an example, presented its characterization in the manuscript and SI. However: 1) the ¹H NMR is not fully assigned, which might lead to wrongful interpretations! How should the authors assign the peaks between 3.2 ppm and 4 ppm and why are there multiple peaks? 2) There are broadened peaks between 2 ppm and 2.7 ppm which are not even labeled/assigned but they still peak a range for integration, and this should be clarified. 3) In Figure 2C, the authors labeled 1-H and 2-H, they should also assign other peaks, including the CH₂ that is in the alpha-position of the ester, the benzyl CH₂, etc.

One more reason why more rigorous characterization is needed is that the authors should make sure there is no consecutive acrylate insertions. It is very common for the acrylate to be polymerized via a radical or anionic pathway.

Please provide detailed formulations for the LOF calculations for all the acrylates mentioned in the work.

Figure 4A on Page 16: the authors mentioned ref 16, which is a review paper. However, there is no mention of the reaction (between dioxane, PEG and 22) or the yields. Please check!

The point-to-point responses

Re: Reviewer #1

This communication deals with the functionalization of polyethylene glycol with a series of acrylate substrates by a photocatalytic process involving an iridium photocatalyst and quinuclidine. The synthetic results obtained and described in Table 1 are interesting but the manuscript has a number of severe issues... Here below, I will list some of these problems in order to be as constructive as possible for the authors.

First of all, we would like to acknowledge reviewer 1's kind efforts in analyzing our manuscript and providing all these constructive suggestions.

Original Comment: 1) First of all the language. I don't speak about English, that can be improve, but about the tones used by the authors. A research publication is not a publicity flyer. Along the manuscript one finds sentences in italics like "...the first example on controllable....." or "...first example on the ammonium ion recognition...". Every time that an interesting result is published, it is the first time it is observed, otherwise it is not so interesting. There is no needing to sell anything, thus more moderate tones are required. Also term as "excellent" should be avoided as well as "we rigorously characterized.....". A rigorous characterization is taken for granted.

Response: Thank you for this professional suggestion. The tones of the manuscript have been revised to be moderate. The highlighted words such as "first example", "excellent", and "rigorously" have been deleted or revised.

Original Comment: 2) While the synthetic part of the paper has a certain degree of interest and may be published in a more specialized journal, the second part of the paper, which occupies more or less half of the manuscript is really wanting. The second section is intitled Mechanism. The study of a reaction mechanism necessary has to involve kinetic measurements. In silico chemistry with imposed reaction trajectories may serve to explain experimental results and cannot substitute an experimental investigation based on kinetic measurements (kinetic constants for a related kinetic scheme). I see that, in the last period, such a publication scheme (mechanisms only based on in silico calculations) is becoming a fashion, but it doesn't bring anywhere.

Response: Thank you for the careful reading and suggestion on the kinetic measurements.

First, we would like to say that the synthetic part of this paper is also very attractive and worthy of publishing. Here are reasons:

- 1) The C-H bond functionalization of PEG is long-standing challenge, particular using acrylates as coupling partners. Different from For's work (*J. Am. Chem. Soc.* **142**, 4581-4585 (2020)) on PEG-initiated polymerization of acrylates, in this work,

we present an efficient C-H bond functionalization of PEG using a series of acrylates, which provides novelty.

- 2) Different from small molecules, one key challenge during the functionalization of PEG is the potential polymer degradation due to the competing C–O bond cleavage (for instant, see *J. Polym. Sci., Part A: Polym. Chem.* **46**, 7386-7394 (2008), *Macromolecules* **43**, 9588-9590 (2010), and *Macromol. Rapid Commun.* **37**, 1587-1592 (2016)). Only one example does not observe significant polymer chain degradation via iron catalysis (*J. Am. Chem. Soc.* **145**, 7612-7620 (2023)) by using strong electron-deficient coupling partners. In this work, we present a new catalytic method for efficient functionalization of PEG, and no polymer chain degradation observing, which provides novelty.

B

D

- 3) The conditions used in this work is not suitable for the similar small molecules, such as dioxane. The key reason is that the noncovalent interaction between polymer chain and quinuclidine-H⁺. This phenomenon is pretty attractive, which also provides novelty.
- 4) The PEGylation of pharmaceuticals is significant important because PEG is one of the ideal drug delivery templates due to hydrophilic and nontoxic properties, which might improve stability and pharmacokinetic properties of drugs. This method is quite efficient and suitable for PEGylation of pharmaceuticals, and we succeeded to achieve one-step PEGylation of bioactive molecules, which provides novelty.

Second, we really appreciate the suggestion on the kinetic measurements. And following this great suggestion, the detailed kinetic experiments of C–H bond alkylation of PEGs by co-catalysis of iridium and quinuclidine have been examined and all the kinetic orders have been determined.

The kinetic orders of the C–H alkylation reaction of PEGs in PEG2000, acrylate **1**, catalyst **PC-1**, and quinuclidine were determined by the method of Variable Time Normalization Analysis (VTNA) reported by Burés et al. (*Angew. Chem. Int. Ed.* **2016**, *55*, 16084–16087). Kinetic orders were determined via inspection of product formation curves when modifying the power (α) of the concentration-adjusted x-axis ($\Sigma[A]^\alpha \Delta t$) to account for the influence of a given reaction component (A) on the overall rate; the kinetic order that provides the best overlap of the product formation curves indicates the reaction order.

To a 4 mL vial were added **PC-1**, quinuclidine, PEG 2000, **1**, and MeCN (4.0 mL) in an N_2 glovebox. The vial was then sealed and transferred out of the glovebox. Under irradiation at 460 nm LEDs, the resulting mixture was stirred for 1, 2, 4, 6, 8, 10 or 12 hours at rt. Evaporation afforded crude **PEG 2000-1**. The level of functionalization was determined by $^1\text{H NMR}$, CH_2Br_2 as internal standard.

1) The data suggests the reaction has **0.5 order** in PEG 2000

2) The data suggests the reaction has *0.3 order* in acylate 1

3) The data suggests the reaction has *0 order* in PC-1

4) The data suggests the reaction has *0 order* in quinuclidine

To sum up, variable time normalization kinetic analysis revealed that under our condition, the reaction rate depend on the concentration of PEG and acrylate but not Ir and quinuclidine catalyst.

All these results have been added in the manuscript and SI.

Original Comment: 3) I do not see any really stabilizing interaction between 18C6 (and PEG) and quinuclidinium ion. The authors cite ref 37 to illustrate the binding between tertiary ammonium ions and crown ethers. In the cited article by Izatt and co-workers, in Table 1, concerning the binding reaction between quinuclidinium iodide (compound 5) and 18C6 in methanol, I read (note d): "Heat produced in these reactions is so small that logK and ΔH cannot be calculated, leading to the conclusion that ΔH and / or logK is very small". From my personal experience, even dialkylammonium ion have very low affinity for 18C6. Thus, the argument presented at page 18, based on the pH of different solutions of quinuclidinium trifluoroacetate in the presence of the different ethers is too weak to demonstrate the point.

Response: Nice concern.

To further evidencing the interaction between 18-C-6 (and PEG) and quinuclidinium ion, a series of experiments were further conducted.

First, the high-resolution mass spectra (HRMS) of a mixture of 18-crown-6 and quinuclidinium ion was examined. Regretfully, there is no peak matching with [18-C-6/ quinuclidinium] ion (desired MW = 376), while quinuclidinium ion (calcd for $C_7H_{14}N^+$ 112.1121, found 112.1129), [18-crown-6 + Na^+] (calcd for $C_{12}H_{24}O_6Na^+$ 287.1465, found 287.1478), and [18-crown-6 + H^+] (calcd for $C_{12}H_{25}O_6^+$ 265.1646, found 265.1658) could be observed. This result indicated that the interaction between quinuclidinium ion and crown ether is indeed weak, which is quite compatible with the conclusion of that "logK is very small".

Item name: ZZN9-28
Item description:

Channel name: 2: Average Time 0.6536 min : TOF MS (50-2000) 6eV ESI+ : Centroided : Combined

1.31e7

Interestingly, as a parallel experiment, the HRMS of crown ether under the same condition observed only [18-crown-6 + Na⁺] (calcd for C₁₂H₂₄O₆Na⁺ 287.1465, found 287.1489) and [18-crown-6 + K⁺] (calcd for C₁₂H₂₄NK⁺ 303.1204, found 303.1225), while [18-crown-6 + H⁺] is barely observed.

Although these experiments can not observe obvious interaction between quinuclidinium ion and crown ether, the interaction between crown ether and proton is indeed proven.

Item name: 9-28-1
Item description:

Channel name: 2: Average Time 0.1450 min : TOF MS (50-2000) 6eV ESI+ : Centroided : Combined

1.31e8

Second, that "logK is very small" is true. However, it is reasonable to predict that the interaction between [quinuclidine-H⁺](OTFA)⁻ and PEG2000 or crown ether would enhance along with the increase of the relative amount of crown ether. Therefore, the

experiments using different ratio of [quinuclidine-H]⁺(OTFA)⁻ and PEG2000 or 18-C-6 were conducted. And chemical shift value of N-H species indeed varied. **Along with the increase of amount of PEG2000 or 18-C-6, the proton at N-H bond would shifted to the downfield of the spectra, indicating that interaction between [quinuclidine-H]⁺(OTFA)⁻ and PEG2000 or 18-C-6 was indeed existed and could be enhanced.**

This conclusion and protocol could be also proven by the known literature, such as *J. Am. Chem. Soc.* **2019**, *141*, 8868–8876.

Third, the low temperature ¹H NMR spectra of [quinuclidine-H]⁺(OTFA)⁻ and [quinuclidine-H]⁺(OTFA)⁻ with PEG 2000 were collected.

The ionization ability of [quinuclidine-H]⁺(OTFA)⁻ decreased gradually along with the temperature decreased. And the chemical shift of N-H bond of [quinuclidine-H]⁺(OTFA)⁻ would shift to high field, which were recorded to be 12.17 (25 °C), 12.05 (0 °C), 11.99 (-10 °C), and 11.95 (-20 °C) ppm.

Alternatively, the chemical shift of N-H bond of [quinuclidine-H]⁺(OTFA)⁻ in the presence of PEG2000 were recorded as 12.36 (25 °C), 12.31 (0 °C), 12.29 (-10 °C), and 12.26 (-20 °C) ppm.

These results indicated that,

- (1) PEG2000 would indeed increase the ionization ability of [quinuclidine-H]⁺(OTFA)⁻ significantly, because the chemical shift value in the presence of PEG2000 at even -20 °C (12.26) is still larger than that in the absence of PEG2000 at room temperature (12.17).
- (2) The temperature decrease affects the ionization ability of [quinuclidine-H]⁺(OTFA)⁻ significantly, because the chemical shift value decreased from 12.36 (25 °C) to 11.95 (-20 °C). The difference is 0.41 ppm. Alternatively, a smaller difference (0.10 ppm) is observed in the presence of PEG2000, which suggests the H-bond between PEG2000 and [quinuclidine-H]⁺(OTFA)⁻ might stabilize the proton significantly, resulting in that temperature decrease has less effect on the ionization ability of [quinuclidine-H]⁺(OTFA)⁻.
- (3) The difference of chemical shift values between [quinuclidine-H]⁺(OTFA)⁻/PEG2000 and [quinuclidine-H]⁺(OTFA)⁻ increases along with the temperature decrease, which also supports the conclusion above.

Based on experiments above, we still believe that **“the interaction between quinuclidinium ion and crown ether or polyether is weak, but it does exist”**.

On the other hand, to address the concern if pH values affect the reaction efficiencies, a series of experiments were designed and conducted.

First, crown ethers, from 12-crown-4 to 24-crown-8, with diverse ring sizes are utilized to react with acrylate **1** under the standard conditions. All the crown ethers could lead to the desired C-H functionalization products, and unexpectedly, 24-C-8 obtained the highest level of functionalization (Fig. a, below). This result indicated that the reaction efficiency might be not affected by pH values since the mixture of [quinuclidine-H]⁺(OTFA)⁻ and 24-C-8 presented a higher pH value than that of 18-C-6 (Fig. b, below). We have revised the description in the main text.

Notably, the level of functionalization increases along with the increase of the number of coordination site (O-atom), which might be responded to the strength of potential H-bonding caused by supermolecular recognition. We envisioned that the reaction efficiency might be affected by the coordination-then-stabilization of [quinuclidine-H]⁺ species with multi- or poly-ethers. Therefore, a series of linear multi-ethers were examined with acrylate **1** (Fig. c, below). And gladly, similar phenomenon was observed, and reaction efficiencies does increase along with the increase of the number of coordination site (O-atom). We concluded that **“supermolecular recognition of multi- or poly-O-atoms with [quinuclidine-H]⁺ would beneficial for proton stabilization and subsequent proton transfer, which would be responded to the increase of reaction efficiencies”**.

Notably, this conclusion could be also supported by a well-defined literature (*J. Am. Chem. Soc.* **1984**, *106*, 510-516), in which the protonation of polyether, glymes, and crown ethers in the gas phase was studied. And the author found that the energy of stabilization of proton did increase along with the increase of -OCH₂CH₂- groups. And interestingly, the energy trend (see below) is very matching with the reaction efficiencies we observed.

Reference: *J. Am. Chem. Soc.* **1984**, *106*, 510-516

Based on the updated experiments, a more reasonable reaction mechanism has been proposed. Firstly, the C-H bond abstraction of PEG with quinuclidine cationic radical via HAT pathway could generate carbon radical and quinuclidine-H⁺ species. The interaction between quinuclidine-H⁺ and poly-coordination of PEG chain stabilizes the proton and enables the formation of **INT2**. The single electron reduction with photocatalyst and subsequent nucleophilic addition with acrylate **1** can produce **INT3**, which undergoes a “formal intramolecular” proton transfer to yield the final product.

Re: Reviewer #2

In this manuscript, the authors successfully established an interesting method to realize C-H bond alkylation of PEGs by co-catalysis of iridium and quinuclidine without observing their significant degradation. A series of acrylate derivatives were suitable for this postmodification. Based on the DFT calculations and detailed mechanistic experiments, the mechanism involving the hydrogen atom abstract, direct reduction of carbon radical, nucleophilic addition, and subsequent protonation was disclosed. This work will provide an alternative for PEGs modification and expand their applications in diverse areas. The manuscript is recommended acceptance after addressing following issues.

We first appreciate reviewer 2's time and efforts on examining this manuscript and providing all these evaluable and constructive suggestions in great details.

Original Comment: 1. Has the equivalence ratio of the catalyst been optimized? Because dual catalysis is used in this work, the proportion of catalyst dosage is mostly likely to affect the reaction efficiency.

Response: Nice concern.

Following the suggestion, the equivalence ratio of catalyst has been screened and we found that increasing the concentration of Ir or quinuclidine did not affect the reaction efficiency. Furthermore, the detailed kinetic experiments also indicated that the reaction has **0 order** in Ir and quinuclidine.

And all these results have been added in the manuscript or SI.

Entry	PC-1 (mol%)	LOF (mol%)
1	0.1	8.3
2	0.2	8.2
3	0.3	8.9
4	0.4	8.7

Entry	quinuclidine (mol%)	LOF (mol%)
1	1.0	8.8
2	2.0	8.3
3	3.0	8.5
4	4.0	8.8

The kinetic orders of the C-H alkylation reaction of PEGs in PEG2000, acrylate **1**, catalyst **PC-1**, and quinuclidine were determined by the method of Variable Time Normalization Analysis (VTNA) reported by Burés et al. (*Angew. Chem. Int. Ed.* **2016**, *55*, 16084–16087). Kinetic orders were determined via inspection of product formation curves when modifying the power (α) of the concentration-adjusted x-axis ($\Sigma[A]^\alpha \Delta t$) to account for the influence of a given reaction component (A) on the overall rate; the kinetic order that provides the best overlap of the product formation curves indicates the reaction order.

To a 4 mL vial were added **PC-1**, quinuclidine, PEG 2000, **1**, and MeCN (4.0 mL) in an N₂ glovebox. The vial was then sealed and transferred out of the glovebox. Under irradiation at 460 nm LEDs, the resulting mixture was stirred for 1, 2, 4, 6, 8, 10 or 12 hours at rt. Evaporation afforded crude **PEG 2000-1**. The level of functionalization was determined by ¹H NMR, CH₂Br₂ as internal standard.

1) The data suggests the reaction has **0.5 order** in PEG 2000

2) The data suggests the reaction has **0.3 order** in acrylate **1**

3) The data suggests the reaction has *0* order in PC-1

4) The data suggests the reaction has *0* order in quinuclidine

quinuclidine

To sum up, variable time normalization kinetic analysis revealed that under our condition, the reaction rate depend on the concentration of PEG and acrylate but not Ir and quinuclidine catalyst.

All these results have been added in the manuscript and SI.

Original Comment: 2. Although a series of acrylate derivatives were attempted, the conversion rate was not considered high. Based on the speculated mechanism, it is most likely that the efficiency of photocatalytic redox is insufficient. Please provide a reasonable explanation and prove that the catalytic system used in the work is sufficiently successful.

Response: Thank you for the comment.

That the conversion was not high is because we did not allow the full conversion of acrylates. Actually, we have monitored the reaction of C-H bond alkylation of PEGs from 0 to 48 hours, and a high conversion rate up to 88% (LOF = 17.6 mol%) could be obtained by prolonging the reaction time. However, the longer reaction time might significantly decrease the concentration of acrylate and subsequently result in PEG chain degradation. Therefore, the reaction time was controlled to make sure that the C-H functionalization could be achieved, at the meanwhile, the PEG chain cleavage could be avoided.

Notably, 1.0-2.0 mol% LOF is typically enough for the further applications of PEG, therefore, the longer reaction time is not necessary for C-H bond modification of polymers.

Time (h)	LOF (mol%)
0	0
1	0.8
2	1.4
5	5.5
12	8.3
24	12.5
48	17.6

Original Comment: 3. The structure of crown ethers will facilitate the Michael addition of acrylates. However, the formation of PEG crown ethers will be destroyed gradually with the increase of the level of functionalization, and the reaction efficiency will become worse and worse, which is an undeniable reason for limiting C-H bond alkylation of PEGs. The authors can further improve the relevant catalytic mechanism, perhaps to obtain better reaction results.

Response: Thank you for the constructive suggestion. And at this stage, it is still challenging and difficult to clarify all the detail during the transformation.

First, we would like to say that the interaction between PEG chain with quinuclidine- H^+ is a dynamic equilibrium process. A formal PEG-based crown ether might be formed to interact with quinuclidine- H^+ , however, this interaction might be a transient process, which could "break and form" continuously. It is very difficult to control the specific O-atom to participate in such an interaction.

Second, less than 20 mol% of acrylate was used to react with PEG, and typically, the LOF was less than 10 mol%. The incorporation of such small LOF might not affect the reaction efficiency significantly. Notably, the time-dependent experiment of PEG 2000 with 1 indeed observed the decreasing of the reaction efficiency along with the reaction going, however, we must say that it might be caused by the decreasing of the concentration of acrylate. We still cannot clarify that that the installation of functional groups into the polymer chain with the increase of LOF destroys the interaction with quinuclidine- H^+ gradually.

Finally, the model reaction of 18-crown-6 and acrylate **1** disclosed that incorporation of multiple functional groups would be feasible since the corresponding multifunctionalized product could be obtained. This result suggests that the increase of the level of functionalization might be not a huge influence to affect the subsequent C-H functionalization.

Original Comment: 4. The energy calculations for INT1A and INT2B seem to be incorrect. Based on the general understanding, ester carbonyl groups can be conjugated with carbon radicals to stabilize molecular configurations, therefore INT2B should have lower molecular energy.

Response: We have re-conducted DFT calculation using 2,5,8,11,14,17-hexaoxaoctadecane (named as PEG260) as starting material.

Due to the conjugation effect of the ester group, the relative energy of INT2B is lower than INT1A. Notably, intermediate INT1A needs to go through transition state TS2B with

an energy of 13.2 kcal mol⁻¹ to result in the formation of more stable intermediate **INT2B**. The energy for overcoming **TS2B** is relatively higher than that of **TS2A**.

Original Comment: 5. Lines 325-330 of page 17, the authors claimed that “In contrast, the relatively lower basicity of the carbanion from 2-alkylmalononitrile might lead to the slower protonation, to favorably protonate carbanion adjacent to nitrile groups.” Can the Michael addition of 2-benzylidenemalononitrile be achieved by replacing quinuclidine with a less alkaline tertiary amine? Please provide reasonable analysis and reasons.

Response: Thank you for the constructive suggestion.

Following the suggestion, we have re-examined other tertiary amines with different alkaline. Interestingly, compared with quinuclidine **B1**, DABCO (1,4-diazabicyclo[2.2.2]octane, **B2**), triethylamine **B3**, and dimethylbenzylamine **B4** can observe the increase of LOF.

Notably, these results are compatible with the acidities of the corresponding conjugate acids (see below, details see *J. Org. Chem.* 1996, 61, 4778–4783). The stronger conjugate acids used, the higher LOF values produced. These results are also compatible with the conclusion of “the relatively lower basicity of the carbanion from 2-alkylmalononitrile might lead to the slower protonation, to favorably protonate carbanion adjacent to nitrile groups.”

These results and discussions have been added in SI.

Re: Reviewer #3

The authors reported C-H functionalization of PEGs, which is enabled by in-situ noncovalent interaction of ammonium ion. This work is based on their recent publication (Controllable C-H Alkylation of Polyethers via Iron Photocatalysis, *J. Am. Chem. Soc.* 2023, 145, 13, 7612–7620), but presented much broader substrate scope and a new catalytic system. This may be a significant breakthrough, but the novelty of this work should be emphasized with greater clarity. ...

First of all, we would like to acknowledge reviewer 3's efforts in analyzing our manuscript.

And we would like to say that this paper including the synthetic part and the novel mechanism part is very attractive and worthy of publishing. Here are reasons:

- 1) The C-H bond functionalization of PEG is long-standing challenge, particular using acrylates as coupling partners. Different from For's work (*J. Am. Chem. Soc.* **142**, 4581–4585 (2020)) on PEG-initiated polymerization of acrylates, in this work, we present an efficient C-H bond functionalization of PEG using a series of acrylates, which provides novelty.
- 2) Different from small molecules, one key challenge during the functionalization of PEG is the potential polymer degradation due to the competing C–O bond cleavage (for instant, see *J. Polym. Sci., Part A: Polym. Chem.* **46**, 7386–7394 (2008), *Macromolecules* **43**, 9588–9590 (2010), and *Macromol. Rapid Commun.* **37**, 1587–1592 (2016)). Only one example does not observe significant polymer chain degradation via iron catalysis (*J. Am. Chem. Soc.* **145**, 7612–7620 (2023)) by using strong electron-deficient coupling partners. In this work, we present a new catalytic method for efficient functionalization of PEG, and no polymer chain degradation observing, which provides novelty.

B

D

- 3) The conditions used in this work is not suitable for the similar small molecules, such as dioxane. The key reason is that the supramolecular interaction between polymer chain and quinuclidine- H^+ would enhance the acidity of quinuclidine- H^+

and facilitate the transformation. This phenomenon is pretty novel and attractive, which also provides novelty.

- 4) The PEGylation of pharmaceuticals is significant important because PEGs is one of the ideal drug delivery templates due to hydrophilic and nontoxic properties, which might improve stability and pharmacokinetic properties of drugs. This method is quite efficient and suitable for PEGylation of pharmaceuticals and even proteins, and we succeeded to achieve one-step PEGylation of bioactive molecules, which provides novelty.

Access to

PEGylation of pharmaceuticals

multifunctional polymeric carrier template

modification of protein

- 5) More mechanistic studies, particularly in the kinetic orders of the C-H alkylation reaction of PEGs in PEG2000, acrylate **1**, catalyst **PC-1**, and quinuclidine were conducted. And the reaction orders are determined to achieve better understanding in the mechanism.
- 6) The mechanistic studies and DFT calculation have arisen a novel mechanism involving the supermolecular interaction and the unexpected radical-reduction – nucleophilic-attack process, which provides novelty.

In sum, we believe this paper is very worth of publishing in Nature Communications.

Original Comment: The characterizations of polymer products are not enough, which might lead to compromised conclusions. The authors used the product PEG 2000-1 as an example, presented its characterization in the manuscript and SI. However: **1)** the ¹H NMR is not fully assigned, which might lead to wrongful interpretations! How should the authors assign the peaks between 3.2 ppm and 4 ppm and why are there multiple peaks? **2)** There are broadened peaks between 2 ppm and 2.7 ppm which are not even labeled/assigned but they still peak a range for integration, and this should be clarified. **3)** In Figure 2C, the authors labeled 1-H and 2-H, they should also assign other peaks, including the CH₂ that is in the alfa-position of the ester, the benzyl CH₂, etc.

Response: Thank you for the careful reading and suggestion.

First, to address concern **1**), the peaks between 3.2 ppm and 4 ppm in ^1H NMR are assigned to be CH_2 or CH of the polymeric PEG chain. The multiple peaks are obtained because of

- The existence of diastereomers.
- The reasonable distribution of level of functionalization by incorporation different equivalence of functional groups into the polymer chain.

PEG chain diastereoisomer
 Two polar groups: 2 diastereomers
 Three polar groups: 4 diastereomers
 Four polar groups: 8 diastereomers
 Five polar groups: 16 diastereomers

Second, to address concern **2**), the peaks between 1.6 and 2 ppm ^1H NMR have been assigned to be CH_2 group at the β -position of the ester group, and the peaks between 2.2 and 2.7 ppm in ^1H NMR have been assigned to be CH_2 groups at the α -position of the ester group.

^1H NMR (400 MHz, CDCl_3) of PEG 2000-1

In addition, to confirm the peak assignment and address concern **3**), the 2D spectrum in Figure 2C has been re-assigned carefully as below. The ^1H - ^1H COSY spectrum clearly showed that the interaction between CH_2 (pink, β -position of ester) and CH_2 (green, α -position of ester), CH_2 (pink, β -position of ester) and $-\text{CHOR}$ (blue). There is no interaction between CH_2 (green, α -position of ester) and $-\text{CHOR}$ (blue) being observed.

All these peak assignments have been added into the manuscript and SI.

Original Comment: One more reason why more rigorous characterization is needed is that the authors should make sure there is no consecutive acrylate insertions. It is very common for the acrylate to be polymerized via a radical or anionic pathway.

Response: Nice concern.

First, the NMR spectra of our product and the consecutive-acrylate-insertion product had been compared carefully, since the later ones have been well characterized by Fors' group in 2020 (*J. Am. Chem. Soc.* **2020**, *142*, 4581–4585). The NMR spectra observed in our work is significantly different from that of the consecutive-acrylate-insertion products. Our assignments are compatible with our spectra.

- 1) Once the consecutive-acrylate-insertion occurs, only one methine group adjunct to O atom forms, which is significantly less than the C-H bonds in the chain. Therefore, there is only a huge singlet peak would be observed and assigned to the PEG chain in ¹H NMR (see below). Alternatively, as we have discussed above, multiple peaks between 3.2 ppm and 4 ppm would be observed in ¹H NMR (see below), which could be caused by the existing of diastereomers and the potential LOF distribution.
- 2) Similar phenomenon was observed in ¹³C NMR spectra. There is only one major carbon peak assigned to CH₂O for PEG chain (see below), alternatively, there are several peaks could be assigned to CH₂O or CHO (see below), which could be caused by the existing of diastereomers and the potential LOF distribution.
- 3) The proton number ratios of C-H at β-position of ester and C-H at α-position of ester for compound PEG2000-1 and PEG2000-2 are very close to 1/1 in our work (see above and below), which are more proper to be assigned as multiple functionalization products instead of consecutive-acrylate-insertion products.

Figure S50. ¹³C NMR of poly(methyl acrylate)-g-poly(ethylene glycol) (Table S6, entry 1).

As comparison, NMR spectra of PEG 2000-2 are shown below.

Moreover, to further confirm the conclusion above, we have also examined the reaction of DME with acrylics **1** as a model reaction under the conditions. The reaction of DME, acrylics **1**, **PC-1**, quinuclidine and MeCN in an N₂ glovebox was conducted under irradiation at 460 nm LEDs for 24 hours at rt. While only one cross-coupling product was formed, we did not observe any homopolymerization or consecutive-acrylate-insertion

products, which could be confirmed by NMR and GC-MS.

In addition, we have also conducted the reaction of 18-crown-6 and acrylate **1**. The major product was mono-substituted 18-crown-6-**1** up to 87% yield. There is no consecutive acrylate insertion product observing.

Based on these two experiments of DME and 18-crown-6 with **1**, we still propose that there is no significant consecutive-acrylate-insertions during our transformation.

Original Comment: Please provide detailed formulations for the LOF calculations for all the acrylates mentioned in the work.

Response: Referring to the previous work (*Chem. Sci.* **2023**, *14*, 9374–9379), the formulation for the LOF calculation is in below and has been added to SI.

The level of functionalization (LOF) = $n(\text{benched alkene in polymer}) / n(\text{monomer of polymer}) = A(\text{benched alkene in polymer}) / A(\text{monomer of polymer})$. n : mole number; A : integral area by ¹H NMR.

Original Comment: Figure 4A on Page 16: the authors mentioned ref 16, which is a review paper. However, there is no mention of the reaction (between dioxane, PEG and 22) or the yields. Please check!

Response: Thank you for the careful reading.

Correctly, ref 16 should be ref 28, and the mistake has been revised in manuscript. In this reference (J. Am. Chem. Soc. 2023, 145, 7612–7620), the Fe-catalyzed C–H alkylation of PEG was reported. In Figure 4, we compared with the reactivity of PEG and benzylidenemalononitrile between Ir/quinuclidine system and such an iron catalysis. Therefore, a reference was added to clarify the iron catalysis used.

Reviewers' Comments:

Reviewer #1:

Remarks to the Author:

Although at this stage, in my view, the reaction mechanism is still far from being fully clarified, since the authors refer to the one proposed as a "plausible mechanism" and since the synthetic results now better appear to be important, I support publication of the manuscript.

Nevertheless, the term "plausible mechanism" should appear also in the conclusion where the adjective "detailed" should be removed (I do not see any fit of the experimental data points with equations related to a precise mechanism scheme).

Furthermore as minor points:

- 1) page 3, line 10, substitute "Stefano" with "Di Stefano"
- 2) page 7, line 4, substitute "well" with "good"
- 3) page 13, line 7 from bottom, substitute "inverted" with another term.

Reviewer #2:

Remarks to the Author:

The authors have fully revised their manuscript and addressed the concerns raised by the reviewers. This reviewer has no further criticism on the revised version. Given the high innovation of the work, this reviewer suggests acceptance this manuscript in current form.

Reviewer #3:

Remarks to the Author:

The authors have addressed the referee comments, and publication is recommended.

The point-to-point responses

Re: Reviewer 1

Although at this stage, in my view, the reaction mechanism is still far from being fully clarified, since the authors refer to the one proposed as a "plausible mechanism" and since the synthetic results now better appear to be important, I support publication of the manuscript.

First of all, we would like to acknowledge reviewer 1's kind efforts in analyzing our manuscript and providing all these constructive suggestions.

Original Comment: Nevertheless, the term "plausible mechanism" should appear also in the conclusion where the adjective "detailed" should be removed (I do not see any fit of the experimental data points with equations related to a precise mechanism scheme).

Response: We have revised "detailed" to "plausible" in manuscript.

Furthermore as minor points:

Original Comment: 1) page 3, line 10, substitute "Stefano" with "Di Stefano"

Response: Thank you for this careful reading. We have revised it.

Original Comment: 2) page 7, line 4, substitute "well" with "good"

Response: We have revised "well" to be "good".

Original Comment: 3) page 13, line 7 from bottom, substitute "inverted" with another term.

Response: We have revised "inverted" to be "distinct".

Re: Reviewer 2

The authors have fully revised their manuscript and addressed the concerns raised by the reviewers. This reviewer has no further criticism on the revised version. Given the high innovation of the work, this reviewer suggests acceptance this manuscript in current form.

We would like to acknowledge reviewer 2's kind approval.

Re: Reviewer 3

The authors have addressed the referee comments, and publication is recommended.

we would like to thank reviewer 3 for the kind approval.